# Increased Serum Thromboxane A2 and Prostacyclin but Lower Complement C3 and C4 Levels in COVID-19: Associations with Chest CT Scan Anomalies and Lowered Peripheral Oxygen Saturation

Hussein Kadhem Al-Hakeim [1,*] , Shaymaa Ali Al-Hamami [2], Abbas F. Almulla [3] and Michael Maes [4,5,6]

1 Department of Chemistry, College of Science, University of Kufa, Najaf 54001, Iraq
2 Department of Medical Laboratory Techniques, Altoosi University College, Najaf 54001, Iraq; shaima.alhmmamy@altoosi.edu.iq
3 Medical Laboratory, Technology Department, College of Medical Technology, The Islamic University, Najaf 54001, Iraq; abbass.chem.almulla1991@gmail.com
4 Department of Psychiatry, Medical University of Plovdiv, 4000 Plovdiv, Bulgaria; dr.michaelmaes@hotmail.com
5 Department of Psychiatry, Faculty of Medicine, Chulalongkorn University, Bangkok 10110, Thailand
6 IMPACT Strategic Research Centre, School of Medicine, Deakin University, P.O. Box 281, Geelong, VIC 3220, Australia
* Correspondence: headm2010@yahoo.com

**Abstract:** COVID-19 patients suffer from hypercoagulation and activated immune-inflammatory pathways. The current study examined the relationship between specific complements and coagulation abnormalities associated with chest CT scan anomalies (CCTAs) and peripheral oxygen saturation (SpO2) in COVID-19 patients. Serum levels of complement C3 and C4, and thromboxane A2 (TxA2) and prostacyclin (PGI2) were measured using an ELISA and albumin, calcium, and magnesium by using the spectrophotometric method in 60 COVID-19 patients and 30 controls. C3 and C4 were significantly decreased ($p < 0.001$), and TxA2 and PGI2 significantly increased ($p < 0.001$) in the COVID-19 patients compared with the controls with the highest levels in the CCTA patients' group. Neural networks showed that a combination of C3, albumin, and TxA2 yielded a predictive accuracy of 100% in detecting COVID-19 patients. SpO2 was significantly decreased in the COVID-19 patients and was inversely associated with TxA2 and PGI2, and positively with C3, C4, albumin, and calcium. Patients with positive IgG results show significantly higher SpO2, TxA2, PGI2, and C4 levels than IgG-negative patients. CCTAs were accompanied by lower SpO2 and albumin and increased PGI2 and TxA2 levels, suggesting that interactions between immune-inflammatory pathways and platelet hyperactivity participate in the pathophysiology of COVID-19 and, consequently, may play a role in an enhanced risk of hypercoagulability and venous thromboembolism. These mechanisms are aggravated by lowered calcium and magnesium levels.

**Keywords:** COVID-19; C3; C4; inflammation; cytokines; biomarkers; thromboxane A2; prostacyclin

## 1. Introduction

The clinical spectrum of SARS-CoV-2 infection ranges from asymptomatic infection to mild upper respiratory tract disease to severe viral pneumonia with respiratory failure and even death [1]. COVID-19 patients frequently suffer from major symptoms, including acute respiratory distress syndrome (ARDS), as well as inflammation with a possible cytokine storm, hypercoagulation, and thrombosis [2–4].

The COVID-19 virus enters the lung cells after binding viral Spike proteins-S with the ACE2 receptors [5] and, consequently, the virus may cause histopathological lesions in the lungs, which appear to be similar to those observed in other forms of ARDS [6].

Coronaviruses activate complement pathways that are a major component of innate immunity and act to remove invading pathogens [7]. Complement activation may result in immune-mediated lung damage [8] and is central to the pathophysiology of lung disorders, including asthma, ARDS [9], and severe COVID-19 disease, which often resembles complementopathies [7]. Activation of the complement system leads to proteolytic cleavage of the key complement molecules C3 and C4 [10], leading to cleavage products including C3a, C3b, C4a, and C4b, which may trigger inflammatory cell recruitment and neutrophil activation [11]. Ghazavi et al. (2020) detected increased C3 and C4 complement levels in non-severe COVID-19 but lower levels in severe COVID-19 patients, which could be explained by increased consumption by forming immune complexes [12]. Low serum C3 levels are detected in critically ill COVID-19 patients and are associated with a poor prognosis [13].

The severity of COVID-19 symptoms, including end-organ damage, is caused by an overzealous inflammatory response, in part, associated with complement activation, endothelial injury, neutrophil activation, thrombophilia, hypercoagulability, and thrombotic microangiopathy [7,14–16]. About one-third of COVID-19 patients in the intensive care unit (ICU) have thrombotic complications, of which venous thromboembolic events are the most common [17]. The association between complement activation and coagulation mechanisms may cause life-threatening complications, and as such, the complement-coagulation network is an important drug target [18]. Nevertheless, only a few studies in COVID-19 focused on C3 and C4 levels in relation to thromboxane A2 (TxA2) and prostacyclin (PGI2).

Endogenous $TxA_2$, which is synthesized from arachidonic acid via cyclooxygenase (COX)-1, COX-2, and TxA2 synthase (TxAS), is produced by activated platelets and exerts prothrombotic effects [19]. TxA2 binds to the prostanoid thromboxane receptor, which triggers the binding to G-proteins, thereby mediating calcium signaling and facilitating platelet aggregation and vasoconstriction [20,21]. COX-1, constitutively expressed in platelets, is a dominant source of TxA2 biosynthesis in humans [22]. In COVID-19, interleukin-1 (IL-1), a pro-inflammatory cytokine, stimulates TxA2 production [23]. PGI2 is mainly produced by endothelial and vascular smooth muscle cells [24] via COX-2 [25]. While TxA2 production causes platelet aggregation and vasoconstriction, PGI2 inhibits platelet aggregation and induces vascular smooth muscle relaxation and endothelium-related vasodilation [26–28]. The endothelial dysfunction following SARS-CoV-2 infection may be caused by lowered endothelial nitric oxide synthase activity and nitric oxide production and VEGF release following systemic hypoxia, while PGI2 may enhance angiogenesis and tissue repair through increased VEGF [29,30].

Recently, we found that chest CT abnormalities (CCTAs) (comprising ground-glass opacities (GGOs), pulmonary densification areas consistent with residual lesions, pneumonic consolidation, and crazy-paving patterns) could be observed in 78.3% of RT-PCR test-positive COVID-19 cases and that the presence of CCTAs was characterized by significantly lowered peripheral oxygen saturation (SpO2) and serum levels of albumin [31]. The latter is a negative acute-phase protein that is lowered in response to the systemic inflammatory response in COVID-19 [1,31–33]. Moreover, lowered SpO2 values were significantly associated with signs of immune activation and positively with albumin, magnesium, and calcium [31]. In addition, the latter study found that lowered serum calcium was the single best biomarker of acute COVID-19 and was more important than inflammatory biomarkers, including interleukin-6 (IL-6) and C-reactive protein (CRP) in discriminating COVID-19 patients from healthy controls. We have argued that beta coronavirus-mediated calcium dyshomeostasis is due to (a) hypoalbuminemia with around 45% of calcium being bound to albumin [34]; and (b) to the activation of store-operated calcium entry (SOCE) channels by endoplasmic-reticulum stress [35,36], which is a consequence of infections with those viruses [37,38].

The present study was conducted to examine the associations between immune-inflammatory (as measured with albumin, C3 and C4) and thrombosis-related (TxA2 and PGI2) biomarkers in relation to SpO2 and CCTAs in COVID-19 patients.

## 2. Materials and Methods

### 2.1. Subjects

The present study recruited sixty patients with confirmed SARS-CoV-2 infection and 30 normal controls. The patients were recruited at the Al-Amal Specialized Hospital for Communicable Diseases and Al-Sadr Teaching Hospital in Najaf governorate, Iraq between September and November 2020. The diagnosis of SARS-CoV-2 infection was based on positive test results of COVID-19 nucleic acids by reverse transcription real-time polymerase chain reaction (rRT-PCR) and positive IgM to SARS-CoV-2, and symptoms of acute infectious disease including fever, fatigue, breathing difficulties, cough, and loss of smell and taste. Patients were excluded for the presence of premorbid medical disease including diabetes, liver disease, chronic kidney disease, neurodegenerative and neurologic disorders including multiple sclerosis, and Parkinson's and Alzheimer's diseases.

Chest computed tomography (CT) scans were used to examine chest CT abnormalities (CCTAs), comprising GGOs, pulmonary densification areas consistent with residual lesions, pneumonic consolidation, and crazy-paving patterns [39]. We divided the patient group into those with (COVID + CCTA) and without (COVID-CCTA) CCTAs. The patients were further divided according to the results of immunoglobulins antibodies against SARS-CoV-2 (IgG) into negative-IgG and positive-IgG subgroups to examine the difference in the measured biomarkers between these subgroups. We also recruited 30 healthy controls, age- and sex-matched to the patient groups. All controls were free from any systemic disease. However, as a public method to enhance their immunity against COVID-19 infection, some healthy controls were taking zinc and vitamins C and D.

The IRB of the "University of Kufa" has approved the study (617/2020). All controls and patients gave written informed consent before participation in this study. The study was conducted according to Iraq and international ethics and privacy laws and was conducted ethically in accordance with the World Medical Association Declaration of Helsinki. Furthermore, our IRB follows the International Guideline for Human Research protection as required by the Declaration of Helsinki, The Belmont Report, CIOMS Guideline, and International Conference on Harmonization in Good Clinical Practice (ICH-GCP).

### 2.2. Measurements

Upon admission of the patient into hospital, RT-PCR tests were conducted using the Lyra® Direct SARS-CoV-2 Assay kits (Quidel Corporation, CA, USA) using the Applied Biosystems® QuantStudio™ 5 Real-Time PCR System (Thermo Fisher Scientific; Life Technologies Holdings Pte Ltd., Marsiling Industrial Estate, Singapore). The Lyra® Direct SARS-CoV-2 assay kit is an RT-PCR assay for the qualitative detection of human coronavirus SARS-CoV-2 from viral RNA extracted from nasal, nasopharyngeal, or oropharyngeal swab specimens.

Upon inclusion in the study, fasting blood samples were taken in the early morning directly after awakening. Five milliliters of venous blood samples were drawn and transferred into clean plain tubes. After ten minutes, the clotted blood samples were centrifuged for five minutes at 3000 rpm, and then serum was separated and transported into three new Eppendorf tubes until assay. Hemolyzed samples were rejected. Serum C3, C4, PGI2, and TxA2 were measured using ELISA techniques based on a sandwich technique using ready-for-use kits supplied by Melsin Medical Co (Jilin, China). The inter-assay CV values of all assays were less than 12%. Serum albumin, magnesium, and total calcium were measured spectrophotometrically using kits supplied by Biolabo® (Maizy, France). The procedures were followed exactly without modifications according to the manufacturer's instructions. A qualitative ACON® COVID-19 IgG/IgM rapid test was used to detect IgG and IgM in all subjects' sera. The kits have a sensitivity $\geq$ 99.1% and a specificity $\geq$ of 98.2%.

*2.3. Statistical Analysis*

The group differences among continuous variables were examined using analysis of variance (ANOVA), while associations between nominal variables were checked using analysis of contingency tables ($\chi^2$-test). Pearson's product-moment and Spearman's rank-order correlation coefficients were used to determine the correlations between biomarkers and clinical and cognitive scores. To assess the associations between diagnosis and biomarkers, we used multivariate general linear models (GLM) while adjusting for confounding variables such as tobacco use disorder (TUD), age, body mass index (BMI), and education. Consequently, we used tests for between-subject effects to determine the relationships between diagnosis and the separate biomarkers. The effect size was estimated using partial eta-squared values. We also computed estimated marginal mean (SE) values provided by the GLM analysis and performed protected pairwise comparisons among treatment means. Binary logistic regression analysis was employed to determine the best predictors of COVID-19 versus the control group. Odd's ratios with 95% confidence intervals were computed as well as Nagelkerke values, which were used as pseudo-$R^2$ values. We used multiple regression analysis to delineate the significant biomarkers predicting symptom domains while allowing for the effects of age, gender, and education. All regression analyses were tested for collinearity using tolerance and VIF values. All tests were two-tailed, with a *p* value of 0.05 used to determine statistical significance.

Neural network analysis was conducted with diagnosis (COVID-19 versus controls) as output variables and biomarkers as input variables, as explained previously [40]. In brief, an automated feed-forward architecture, multilayer perceptron neural network model was employed to check the associations between biomarkers (input variables) and the diagnosis of COVID-19 versus controls (output variables). We trained the model with two hidden layers with up to 4 nodes in each layer, 20–50 epochs, and minibatch training with gradient descent. One consecutive step with no further decrease in the error term was used as a stopping rule. We extracted the following three samples: (a) a holdout sample (33.3%) to check the accuracy of the final network, (b) a training sample (47.7%) to estimate the network parameters, and (c) a testing sample (20.0%) to prevent overtraining. We computed error, relative error, and importance and relative importance of all input variables. IBM SPSS windows, Armonk, NY version 25, 2017 was used for all statistical analysis.

## 3. Results

*3.1. Socio-Demographic Data*

Table 1 shows the socio-demographic and clinical data in the COVID-19 patients and the healthy control (HC) group. There was no significant difference between the study groups in age, BMI, education, residency, marital status, and TUD. Sixty patients were recruited to participate, namely, from the admission room: 35 patients, ICU: 16 patients, and RCU: 9 patients. All the patients were on $O_2$ therapy, and were administered paracetamol, bromhexine, vitamin C, vitamin D, and zinc. Thirty-six patients out of 60 had a positive SARS-CoV-2 IgG antibodies test.

**Table 1.** Socio-demographic and clinical data of COVID-19 patients and healthy controls (HC).

| Variables | HC (*n* = 30) | COVID-19 (*n* = 60) | F/FEPT/$\chi^2$ | df | *p* |
|---|---|---|---|---|---|
| Age (years) | 40.1 ± 8.8 | 41.0 ± 10.2 | 0.17 | 1/88 | 0.681 |
| BMI (kg/m$^2$) | 26.05 ± 4.02 | 27.07 ± 3.62 | 1.50 | 1/88 | 0.225 |
| Sex (Female/Male) | 6/24 | 17/43 | 0.73 | 1 | 0.393 |
| Urban/Rural | 28/2 | 52/8 | | | 0.486 |
| Single/married | 10/20 | 17/43 | 0.24 | 1 | 0.626 |

**Table 1.** *Cont.*

| Variables | HC (*n* = 30) | COVID-19 (*n* = 60) | F/FEPT/$\chi^2$ | df | *p* |
|---|---|---|---|---|---|
| TUD (No/Yes) | 20/10 | 39/21 | 0.03 | 1 | 0.875 |
| Employment (No/Yes) | 9/21 | 21/39 | 0.23 | 1 | 0.635 |
| Education (years) | $10.7 \pm 3.2$ | $9.8 \pm 4.0$ | 1.07 | 1/88 | 0.303 |
| Admission room / ICU / RCU | | 35/16 /9 | | | |
| Zinc (No/Yes) | 23/7 | 1/59 | 57.53 | 1 | <0.001 |
| Vitamin D (No/Yes) | 27/3 | 0/60 | 77.14 | 1 | <0.001 |
| Vitamin C (No/Yes) | 23/7 | 0/60 | 61.79 | 1 | <0.001 |
| Dexamethasone (No/Yes) | | 9/51 | | | |
| Azithromycin (No/Yes) | | 25/35 | | | |
| Enoxaparin (No/Yes) | | 28/32 | | | |
| O$_2$ therapy (No/Yes) | | 0/60 | | | |
| Bromhexine (No/Yes) | | 21/39 | | | |
| Paracetamol (No/Yes) | 27/3 | 0/60 | 77.14 | 1 | <0.001 |
| Omeprazole (No/Yes) | | 23/37 | | | |
| Ceftriaxone (No/Yes) | | 35/25 | | | |
| IgG (Positive / Negative) | | 36/24 | | | |

Results are shown as mean $\pm$ SD, FEPT: Fisher's exact probability test, BMI: body mass index, TUD: tobacco use disorder, and immunoglobulins antibodies against SARS-CoV-2.

### 3.2. Differences in Biomarkers among Groups

The results of the biomarkers in the COVID-19 patients compared with the control group are presented in Table 2. There was a significant decrease in SPO$_2$ percentage in the COVID + CCTA patients compared to the COVID-CCTA patients and the controls. There was also a significant decrease in the serum albumin levels in the COVID-19 patients compared to the control group, with the lowest levels in the COVID + CCTA group. Serum magnesium was not significantly different between the three study groups. Serum PGI2 and TxA2 showed a significant increase in the COVID patients compared with the controls, with the highest levels in the COVID + CCTA group. The significant increases in TxA2 in COVID-19 remained significant after adjustment for albumin (F = 6.93, df = 2/86, *p* = 0.002), and prostacyclin (F = 3.33, df-2/85, *p* = 0.040). Serum C3, C4, and total calcium were significantly lower in both patient groups when compared with the controls. The differences in serum total calcium were no longer significant after covarying for albumin levels (F = 0.16, df = 1/86, *p* = 0.850).

### 3.3. Multivariate Differences between COVID-19 Patients and Controls

Using lowered levels of SpO2, C3, and albumin in a binary logistic regression analysis, we found a significant discrimination between the COVID-19 patients and the controls with an accuracy of 100%. The network information of a neural network model that examines the discrimination of the COVID-19 versus the controls is presented in Table 3. The network has been trained using one hidden layer with four units in layer one. A hyperbolic tangent was used as an activation function in hidden layer one and identity in the output layer. The partitioned confusion matrix showed an AUC ROC = 1.000 with an accuracy of 100.0% in the holdout sample and a sensitivity of 100.0% and specificity of 100.0%. Figure 1 displays the relative importance of the most effective input variables (C3, albumin, TxA2, C4, PGI2, and SPO$_2$) that represent the most important determinants of the model's predictive power.

**Table 2.** Biomarkers in COVID-19 patients divided into those with chest CT scan abnormalities (COVID + CCTA) and those without CCTA (COVID-CCTA) and healthy controls (HC).

| Biomarkers | HC [A] (*n* = 30) | COVID-CCTA [B] (*n* = 15) | COVID + CCTA [C] (*n* = 45) | F | df | *p* |
|---|---|---|---|---|---|---|
| SPO$_2$ | 98.53 ± 0.68 [C] | 97.13 ± 0.74 [C] | 92.62 ± 3.40 [A,B] | 56.37 | 2/87 | <0.001 |
| Albumin g/L | 43.10 ± 3.11 [B,C] | 33.47 ± 5.08 [A,C] | 29.71 ± 3.79 [A,B] | 111.12 | 2/87 | <0.001 |
| Magnesium mM | 0.933 ± 0.102 [C] | 1.018 ± 0.223 | 1.010 ± 0.172 [A] | 2.42 | 2/87 | 0.005 |
| Calcium mM | 2.264 ± 0.096 [B,C] | 2.089 ± 0.148 [A] | 2.021 ± 0.128 [A] | 36.02 | 2/87 | <0.001 |
| Thromboxane A2 pg/mL | 236.4 ± 64.6 [B,C] | 337.9 ± 64.1 [A] | 366.0 ± 98.7 [A] | 22.17 | 2/87 | <0.001 |
| Prostacyclin pg/mL | 104.7 ± 36.9 [B,C] | 153.6 ± 45.4 [A,C] | 188.7 ± 45.0 [A,B] | 35.00 | 2/87 | <0.001 |
| Complement C3 mg/L | 872.1 ± 285.5 [B, C] | 346.7 ± 134.0 [A] | 302.9 ± 88.2 [A] | 92.00 | 2/87 | <0.001 |
| Complement C4 mg/L | 381.7 ± 138.7 [B, C] | 266.3 ± 110.0 [A] | 273.1 ± 87.3 [A] | 9.98 | 2/87 | <0.001 |

All results are shown as mean (SE). All results of univariate GLM analyses examining the associations between diagnosis of healthy control and COVID-19 patients divided into two groups depending on the presence or absence of chest CT abnormalities (CCTA) after adjusting for age, sex, tobacco use disorder, and body mass index. [A,B,C]: pair wise comparisons, SPO$_2$ %: oxygen saturation percentage.

**Table 3.** Results of neural networks with diagnosis of COVID-19 versus heathy controls (HC) as output variables and biomarkers as input variables.

| | Model | COVID-19 vs. HC |
|---|---|---|
| Input Layer | No. of units | 6 |
| | Rescaling method | Normalized |
| Hidden layers | No. of hidden layers | 1 |
| | No. of units in hidden layer 1 | 4 |
| | Activation Function | Hyperbolic tangent |
| Output layer | Dependent variables | COVID-19 vs. HC |
| | Number of units | 2 |
| | Activation function | Identity |
| | Error function | Sum of squares |
| Training | Sum of squares error term | 0.420 |
| | Incorrect or relative error % | 0.0% |
| | Prediction (sens, spec) | 100%, 100% |
| Testing | Sum of Squares error | 0.252 |
| | Incorrect or relative error % | 0.0% |
| | Prediction (sens spec) | 100%, 100% |
| | AUC ROC | 1.00 |
| Holdout | Incorrect or relative error % | 0.0% |
| | Prediction (sens, spec) | 100%, 100% |

AUC ROC: area under curve of receiver operating curve, sens: sensitivity, spec: specificity.

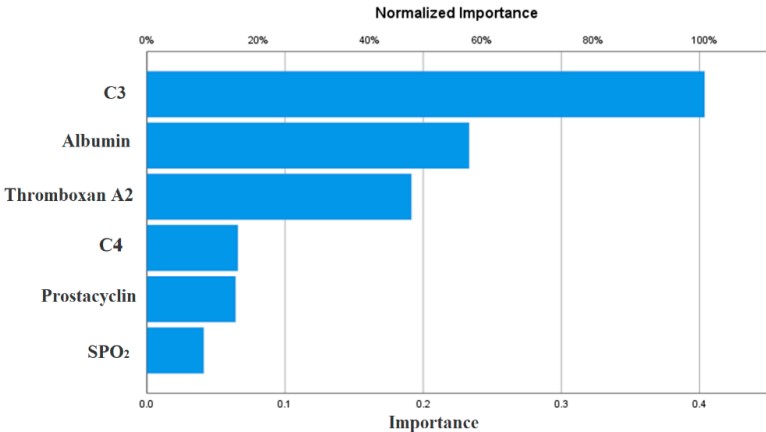

**Figure 1.** Results of neural network (importance chart) with diagnosis of COVID-19 as output variables and biomarkers (in z-scores) as input variables.

### 3.4. Associations of Biomarkers with Anti-SARS-CoV-2 IgG Antibodies, CCTAs, and SpO2

The biomarkers in the COVID-19 patients with positive versus negative anti-SARS-CoV-2 IgG antibody titers are presented in Table 4. The positive IgG group showed a significant increase in $SpO_2$, TxA2, PGI2, and C4 compared with the patients with negative IgG antibodies. These differences remained significant after FDR *p*-correction (at *p* = 0.016). The same table shows that $SpO_2$ and albumin were significantly lower and prostacyclin significantly higher in the COVID + CCTA as compared with the COVID-CCTA patients. These differences remained significant after FDR *p*-correction (at *p* = 0.033).

**Table 4.** Differences in biomarkers between COVID-19 patients with and without anti-SARS-CoV-2 IgG antibodies and with and without chest CT scan anomalies (CCTAs).

| Biomarkers | Negative IgG *n* = 24 | Positive IgG *n* = 36 | F | df | *p* |
|---|---|---|---|---|---|
| $SpO_2$ | 92.25 ± 4.31 | 94.75 ± 2.56 | 7.97 | 1/58 | 0.007 |
| Albumin g/L | 29.83 ± 4.37 | 31.19 ± 4.43 | 1.37 | 1/58 | 0.246 |
| Magnesium mM | 1.017 ± 0.201 | 1.009 ± 0.174 | 0.02 | 1/58 | 0.880 |
| Calcium mM | 2.012 ± 0.132 | 2.056 ± 0.137 | 1.54 | 1/58 | 0.219 |
| Thromboxane A2 pg/mL | 306.3 ± 57.6 | 394.1 ± 93.8 | 16.78 | 1/58 | <0.001 |
| Prostacyclin pg/mL | 160.2 ± 40.9 | 193.1 ± 47.2 | 7.78 | 1/58 | 0.007 |
| Complement C3 mg/L | 329.0 ± 120.2 | 303.7 ± 88.4 | 0.88 | 1/58 | 0.351 |
| Complement C4 mg/L | 227.8 ± 81.1 | 300.4 ± 89.2 | 10.23 | 1/58 | 0.002 |
| Biomarkers | No CCTAs *n* = 15 | CCTAs *n* = 45 | F | df | *p* |
| $SpO_2$ | 97.13 ± 0.74 | 92.62 ± 3.40 | 25.94 | 1/58 | <0.001 |
| Albumin g/L | 33.47 ± 5.08 | 29.71 ± 3.79 | 9.25 | 1/58 | 0.004 |
| Magnesium mM | 1.0178 ± 0.233 | 1.010 ± 0.172 | 0.02 | 1/58 | 0.895 |
| Calcium mM | 2.089 ± 0.147 | 2.038 ± 0.135 | 2.94 | 1/58 | 0.092 |
| Thromboxane A2 pg/mL * | 337.9 ± 64.1 | 366.0 ± 98.7 | 1.09 | 1/58 | 0.301 |
| Prostacyclin pg/mL | 153.6 ± 45.4 | 188.7 ± 45.0 | 6.81 | 1/58 | 0.011 |
| Complement C3 mg/L ** | 346.7 ± 134.0 | 302.9 ± 88.2 | 1.92 | 1/58 | 0.171 |
| Complement C4 mg/L | 266.3 ± 110.0 | 273.4 ± 87.3 | 0.59 | 1/58 | 0.809 |

All results of univariate GLM analysis; results are shown as mean ± SE. * Processed in Ln and ** square root transformation. $SpO_2$ %: oxygen saturation percentage.

The intercorrelation matrix between $SpO_2$ and the biomarkers is shown in Table 5. $SpO_2$ was significantly and negatively correlated with TxA2 and prostacyclin, and positively with C3, C4, albumin, and calcium. TxA2 was significantly and positively correlated with prostacyclin and negatively with C3, albumin, and calcium. Prostacyclin levels were significantly and positively correlated with magnesium and inversely with C3, albumin, and calcium.

**Table 5.** Intercorrelation matrix between oxygen saturation percentage ($SpO_2$) and different biomarkers of COVID-19.

| Biomarkers | $SpO_2$ | Thromboxane A2 | Prostacyclin |
|---|---|---|---|
| Thromboxane A2 | −0.362 (0.001) | | |
| Prostacyclin | −0.380 (0.001) | 0.539 (0.002) | |
| Complement C3 | 0.598 (0.001) | −0.544 (0.002) | −0.593 (0.001) |
| Complement C4 | 0.355 (0.001) | 0.028 (0.800) | −0.014 (0.9) |
| Albumin | 0.617 (0.001) | −0.549 (0.002) | −0.572 (0.001) |
| Magnesium | −0.208 (0.055) | 0.178 (0.115) | 0.308 (0.005) |
| Calcium | 0.490 (0.001) | −0.379 (0.002) | −0.496 (0.001) |

All results of partial correlation analysis after covarying for age, sex, body mass index, and tobacco use disorder. Shown are false discovery rate corrected *p* values.

Table 6 shows the regression of TxA2 on immune-inflammatory biomarkers. The first regression shows that 39.3% of the variance in TxA2 was explained by the regression on albumin (inversely) and prostacyclin (positively). The second regression shows that 42.0% of the variance in TxA2 was explained by the regression on C3 (inversely) and C4 and prostacyclin (both positively).

**Table 6.** Results of multiple regression analysis with PxA2 as dependent variables and immune-inflammatory mediators and prostacyclin.

| Dependent Variables | Explanatory Variables | β | t | *p* | F Model | df | *p* | $R^2$ |
|---|---|---|---|---|---|---|---|---|
| **#1.** LnTxA2 | **Model** | | | | 28.128 | 2/87 | <0.001 | 0.393 |
| | Albumin | −0.387 | −3.923 | <0.001 | | | | |
| | Prostacyclin | 0.328 | 3.319 | 0.001 | | | | |
| **#2.** LnTxA2 | **Model** | | | | 20.779 | 3/86 | <0.001 | 0.420 |
| | sqrC3 | −0.525 | −4.471 | <0.001 | | | | |
| | C4 | 0.241 | 2.498 | 0.014 | | | | |
| | Prostacyclin | 0.227 | 2.118 | 0.037 | | | | |

## 4. Discussion

### 4.1. Changes in Complement in COVID-19

The first major finding of the present study is that C3 and C4 are significantly decreased in COVID-19 patients. As reviewed in the introduction, there were some reports that C3 is significantly lowered in severe COVID-19 as compared with controls. Increased cleavage during activation and higher consumption after immune complex production could account for this result [12]. C3 levels tend to increase gradually in recovered COVID-19 patients, whilst C3 levels were decreased in non-survivors and associated with increased risk of in-hospital death [13]. The levels of complement C4 were decreased from day 0 to day 10 in patients hospitalized for more than two weeks, but not in patients who were discharged earlier [41]. In a recent meta-analysis, a strong correlation between COVID-19 severity

and mortality and C3 and C4 contents was found, which indicate reduced complement activation [42]. Furthermore, C3 and C4 may be helpful in identifying patients who are at high risk of negative clinical outcomes [42]. However, in a previous analysis, no major variations in complement C3 or C4 levels were observed between severe and less severe COVID-19 study groups [43], whereas another report found increased C3 and C4 in COVID-19 patients [44].

We also found that lowered SpO2 is associated with lowered C3 and C4 levels. In this respect, systemic complement activation is associated with respiratory failure in COVID-19 patients [45]. Complement activation mediates, in part, the systemic immune-inflammatory response in SARS-CoV infection [8] and the activation of complement C3 can worsen SARS-COV-related ARDS [46].

### 4.2. Increased TxA2 and PGI2 in COVID-19

The second major finding of this study is that TxA2 is significantly increased in COVID-19 patients when compared with controls. Platelets produce significant amounts of TxA2 and prostaglandins dependent upon the activity of COX-1, COX-2, and TxA2. On platelets, TxA2 binds to the prostanoid thromboxane receptor, thereby initiating an amplification loop leading to further platelet activation, aggregation, and TxA2 formation [47], which may, consequently, lead to a prothrombotic state with an increased mortality risk [17,48,49]. Increased platelet activity and aggregability has been reported in patients with COVID-19 [50] and is associated with an increased risk of death [51]. In addition, coagulopathies are often observed in COVID-19 with up to one-third of patients having thrombotic problems [52].

In our study, we observed a significant intertwined upregulation in TxA2 and PGI2 levels. Prostaglandins, including PGI2, are usually raised in response to inflammatory or toxic stimuli [53]. Endothelial PGI2 binds to the Gs-coupled PGI2 receptor on platelets, thereby reducing platelet reactivity, which can be critical to minimizing the risk for atherothrombotic events [54]. PGI2 signaling induces cytosolic cAMP, thereby preventing platelet activation [55] and may reduce viral-induced illness by suppressing the induction of type-I interferons [56]. PGI2 protects against cytokine toxicity by attenuating nuclear factor-κB activity and possesses strong anti-inflammatory and immune-regulatory properties [57]. As such, increased levels of prostacyclin may attenuate the thrombotic and immune effects of increased TxA2. Nevertheless, in our study, we found that the increases in TxA2 in the COVID-19 patients remained significant after adjusting for albumin and prostacyclin. In this regard, it is important to note that PGI2 signaling may lead to an increased production of IL-6 from stromal cells [58] and may promote T helper-1 differentiation possibly through cAMP-PKA signaling [59].

The massive platelet activation in COVID-19 is probably not a direct consequence of the virus itself because SARS-CoV-2 has rarely been found in the serum of infected patients [60]. One of the mechanisms causing severe COVID-19 is believed to stem from an exaggerated immune-inflammatory response with complement-induced-coagulation, massive endothelial damage, and systemic microangiopathy [61]. In severe COVID-19, widespread endothelial dysfunction and coagulopathies and complement-induced thrombosis may cause systemic microangiopathy and thromboembolism, which may lead to multi-organ failure, thereby causing death [61]. Moreover, in COVID-19, alternative complement pathway activation is associated with microvascular injury and thrombosis [62]. Consequently, a pro-coagulative endothelium may induce endothelins, thereby mediating the infiltration of inflammatory cells in the lungs leading to ARDS in COVID-19 [63–65]. On the other hand, the endothelium mediates antithrombotic and anti-inflammatory functions by releasing active endothelium-derived factors such as nitric oxide PGI$_2$ [66], but these regulatory functions are frequently insufficient.

### 4.3. Lowered Albumin, Calcium, and Magnesium in COVID-19

In agreement with Al-Hakeim et al. (2020) and other studies reviewed in the latter paper [31], this study found that serum albumin, calcium, and magnesium were signifi-

cantly lowered in COVID-19. Hypoalbuminemia in infectious disease may be explained by the acute phase responses in COVID-19 with an increased breakdown of albumin and an increased production of positive acute phase proteins [67], and by an enhanced capillary permeability leading to the leaking of albumin to the interstitial space [68]. Interestingly, in the current study, we found significant and inverse associations between TxA2, C3, and albumin levels, suggesting platelet hyperactivity–immune-inflammatory interactions in COVID-19. A previous study showed that hypoalbuminemia, especially when serum albumin is <35 g/L, is associated with elevated D-dimers and an enhanced risk of artery and venous thrombosis [69,70]. The association between hypoalbuminemia and hypercoagulability and venous thromboembolism may be explained by the anticoagulant and antiplatelet activities of albumin [71]. Not only platelet–platelet but also platelet–leukocyte interactions play a key role in COVID-19 [50]. Activation of the prostanoid TxA2 receptor mediates not only thrombosis and angiogenesis, but also vascular inflammation [23]. In ARDS, platelets may function as effectors in immunity and inflammation [72,73] and virus–platelet interactions increase thrombotic risk by fostering procoagulant and inflammatory states during viral infection [74]. The current study found that a combination of C3, albumin, and TxA2 could be used as an external validating criterion for the diagnosis of COVID-19. These data further underscore that the combined intertwined activities of immune-inflammatory and hemostasis pathways underpin the pathophysiology of COVID-19.

Moreover, in agreement with Al-Hakeim et al. (2021), this study found that CCTAs are accompanied by lowered SpO2 and hypoalbuminemia, and additionally, that lowered SpO2 is associated with low serum albumin [31]. Such findings indicate that the lesions caused by inflammatory lung damage may result in diminished lung oxygenation and systemic inflammation with hypoalbuminemia [31]. Interestingly, CCTAs are associated with the severity of COVID-19 [75] and are an important risk factor for myocarditis in COVID-19 patients [75].

It is well-known that both calcium and magnesium are partially bound to albumin and, therefore, hypoalbuminemia may explain at least part of the lowered magnesium and calcium levels in COVID-19 [31]. Importantly, hypocalcemia is more common in severely ill COVID-19 patients than in mild cases, and may detect with high specificity the more critically ill patients or those with a poorer outcome [76,77] and may predict the hospitalization of COVID-19 patients [78].

*4.4. Biomarkers and IgG Positivity*

Another major finding is that the patients with positive IgG titers showed higher SpO2, TxA2, PGI2, and C4 levels. Previously, we reported that the patients with IgG positivity showed higher IL-6, sRAGE, and ACE2 levels as compared to the patients with negative IgG titers [31]. We found 60.0 (the current study) and 66.7% [31] of COVID-19 patients showed positive IgG levels, whereas a previous study reported that 77.9% of COVID-19 patients were IgG positive [79]. The antibody dynamics observed in COVID-19 patients are quite similar to the Ig responses in other viral infections, with an initial increase in IgM and increasing IgG levels when IgM starts to decrease [80]. Not only increased IL-6 may drive B-cell mediated IgG formation [81], but complement activation also promotes humoral immunity [82]. Increased IgG titers are more pronounced in severe than in mild cases [83], and, therefore, the association between positive IgG titers and TxA2 and PGI2 found in the current study could reflect the severity of illness.

The relatively low sample size and the need for measurements of other complements and coagulation related molecules are the main limitations of the present study.

## 5. Conclusions

COVID-19 is characterized by significantly decreased SpO2, C3, and C4 and significantly increased TxA2 and PGI2. A combination of C3, albumin, and TxA2 yielded a predictive accuracy of 100% in discriminating COVID-19 patients from healthy controls. SpO2 was significantly and positively associated with C3, C4, albumin, and calcium, and

negatively with TxA2 and PGI2. Furthermore, CCTAs are accompanied by lower SpO2 and albumin, but higher prostacyclin. The strong association between CCTA-associated hypoalbuminemia and increased TxA2 suggests that intertwined interactions between immune-inflammatory pathways and platelet hyperactivity participate in the pathophysiology of COVID-19. These mechanisms may be aggravated by lowered calcium and magnesium and, consequently, an increased risk of hypercoagulability and venous thromboembolism.

**Author Contributions:** Conceptualization, H.K.A.-H. and M.M.; methodology, S.A.A.-H., H.K.A.-H. and A.F.A.; software, S.A.A.-H.; validation, H.K.A.-H. and M.M.; formal analysis, M.M.; investigation, H.K.A.-H., M.M., S.A.A.-H. and A.F.A.; data curation, H.K.A.-H., M.M., S.A.A.-H. and A.F.A.; writing—original draft preparation, M.M. and H.K.A.-H.; writing—review and editing, M.M. and H.K.A.-H.; supervision, M.M. and H.K.A.-H. All authors have read and agreed to the published version of the manuscript.

**Funding:** This study was not officially funded.

**Institutional Review Board Statement:** The study was approved by The IRB of the University of Kufa under the number 617/2020.

**Informed Consent Statement:** All participants have given informed consent.

**Data Availability Statement:** All data are available in the published manuscript.

**Acknowledgments:** We thank the staff of Al-Sadr Teaching Hospital and Al-Amal Specialized Hospital for Communicable Diseases in Najaf governorate, Iraq for their help in recruiting patients. We thank the high-skilled staff of the Department of Occupational Safety, Health, and the Environment labs of the Imam Ali (PBUH) Holy Shrine for the help in the biomarker assays.

**Conflicts of Interest:** All authors declared no conflict of interest regarding the present research.

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
