# Peer review of "Increased Serum Thromboxane A2 and Prostacyclin but Lower Complement C3 and C4 Levels in COVID-19: Associations with Chest CT Scan Anomalies and Lowered Peripheral Oxygen Saturation"

_covid, doi:10.3390/covid1020042_

Round 1
Reviewer 1 Report
Dear authors
Please consider the following comment:
- Abstract. The abstract should be rewritten in a correct and direct style by using the past, respecting the IMRAD structure,
- "This study was ........ (SpO2).
The aim of the study should be rewritten in a general objective without operational (outcomes) details.
- "C3 and C4 are significantly decreased"
Please use the past (were)
Lines 25-26. Statement regarding hypoalbuminemia are results of the current study? If not, it should be removed.
Numerical values should be added.
Introduction
Line 77. "we published that ...." Replace published by found
Line 80. "PCR test–proven COVID-19" Replace proven by positive
Lines 95-97. The objective may specify the study population and the disease.
Results.
Line 112. COVID-CCLA. This abbreviation is not defined!!
Lines 117-119. "The significant increases i........ p=0.040)." this sentence should be rewritten in a correct and simple style.
Line 128. CCTA should be defined although it has been mentionned in the introduction.
Lines 133-134. "Nevertheless, the SPSS program does not allow to estimate the regression parameters in this condition." Why the authors inserted this sentence?
All the apargaraphs begun with table nµ° shows. It'd be preferable using other sentences (as shown in table; according to our results,......).
Discussion
Line 192. "As reviewed in the introduction, ....." this should be removed. Write directly and add references.
Lines 184-185. "This may ............... formation" This senetnce has been included in the introduction. It should be paraprased or replaced.
Line 194-196. Write in a concise and direct style
Line 268. "As reviewed in our Introduction ..." Avoid such expressions and write directly.
Line 284. "As we discussed previously,..." same previous comment
Limitations of the study should be added at the end of discussion
Add a conclusion
Author Response
Dear Editor.
The response to the Reviewer 1 comments are attached as WORD fiel.
Best regards
Yours
The Corresponding Author

Reviewer 2 Report
The manuscript describes a study regarding inflammatory biomarkers during COVID-19. The authors have found that thromboxane A2 and prostacyclin are increased in COVID-19 patients while C3 and C4 are lowered.
The work provides markers associated with COVID-19 that could be useful for diagnosis and treatment.
I have the following comments that I hope will be helpful in improving the manuscript quality.
1) The abstract describes the association of biomarkers with COVID-19 but does not present their association with CT scan anomalies and SpO2. Hence the title seems disconnected from the abstract.
2) The paper should include the following reference regarding C3 and C4 biomarkers: "Serum Complement C3 and C4 and COVID-19 Severity and Mortality: A Systematic Review and Meta-Analysis With Meta-Regression"
3) Line 40. "the virus may cause histopathological lesions". This statement seems disconnected from the context.
4) Lines 92-94. "Magnesium has antioxidant [39], anti-inflammatory [40] and antithrombotic [41] properties with about one third of total magnesium levels being bound to albumin [42]." How this information is important for the study?
5) Line 105. "Thirty-six patients out of 60 were IgG positive." Please add for SARS-CoV-2.
6) Line 109-122. Please correct "CCLA" with "CCTA". Please explain the abbreviation for positive or negative for CCTA.
7) How the biomarkers analyzed are important in other respiratory infections or diseases?
Author Response
Dear Editor
We responded positively to all Reviewers #2 comments.
Please, find attached our response in detail.
Best regards
Yours
Prof.Dr. Al-Hakeim
The Corresponding Author

Round 2
Reviewer 2 Report
The authors have revised the manuscript accordingly to my suggestions.